# Aspirin-triggered resolvin D1 reduces parasitic cardiac load by decreasing inflammation in a murine model of early chronic Chagas disease

Ileana Carrillo[1], Rayane Aparecida Nonato Rabelo[2], César Barbosa[3], Mariana Rates[3], Sebastián Fuentes-Retamal[1], Fabiola González-Herrera[1], Daniela Guzmán-Rivera[1,4], Helena Quintero[1], Ulrike Kemmerling[5], Christian Castillo[6], Fabiana S. Machado[2,3], Guillermo Díaz-Araya[7]*, Juan D. Maya[1]*

1 Programa de Farmacología Molecular y Clínica, ICBM, Facultad de Medicina, Universidad de Chile, Santiago, Chile, 2 Programa em Ciências da Saúde, Doenças Infecciosas e Medicina Tropical/ Laboratório Interdisciplinar de Investigação Médica, Faculdade de Medicina, Universidade Federal de Minas Gerais, Belo Horizonte, Brazil, 3 Laboratório de Imunorregulação de Doenças Infecciosas, Departamento de Bioquímica e Imunologia, ICB, Universidade Federal de Minas Gerais, Belo Horizonte, Brazil, 4 Escuela de Farmacia, Facultad de Medicina, Universidad Andrés Bello, Santiago, Chile, 5 Programa de Anatomía y Biología del Desarrollo, ICBM, Facultad de Medicina, Universidad de Chile, Santiago, Chile, 6 Núcleo de Investigación Aplicada en Ciencias Veterinarias y Agronómicas, Facultad de Medicina Veterinaria y Agronomía, Universidad de Las Américas, Santiago, Chile, 7 Departamento de Farmacología Química y Toxicología, Facultad de Ciencias Químicas y Farmacéuticas, Universidad de Chile, Santiago, Chile

* gadiaz@ciq.uchile.cl (GD-A); jdmaya@uchile.cl (JDM)

**Data Availability Statement:** All relevant data are within the manuscript and its Supporting Information files.

## Abstract

### Background

Chagas disease, caused by the protozoan *Trypanosoma cruzi*, is endemic in Latin America and is widely distributed worldwide because of migration. In 30% of cases, after years of infection and in the absence of treatment, the disease progresses from an acute asymptomatic phase to a chronic inflammatory cardiomyopathy, leading to heart failure and death. An inadequate balance in the inflammatory response is involved in the progression of chronic Chagas cardiomyopathy. Current therapeutic strategies cannot prevent or reverse the heart damage caused by the parasite. Aspirin-triggered resolvin D1 (AT-RvD1) is a pro-resolving mediator of inflammation that acts through N-formyl peptide receptor 2 (FPR2). AT-RvD1 participates in the modification of cytokine production, inhibition of leukocyte recruitment and efferocytosis, macrophage switching to a nonphlogistic phenotype, and the promotion of healing, thus restoring organ function. In the present study, AT-RvD1 is proposed as a potential therapeutic agent to regulate the pro-inflammatory state during the early chronic phase of Chagas disease.

### Methodology/Principal findings

C57BL/6 wild-type and FPR2 knock-out mice chronically infected with *T. cruzi* were treated for 20 days with 5 µg/kg/day AT-RvD1, 30 mg/kg/day benznidazole, or the combination of 5 µg/kg/day AT-RvD1 and 5 mg/kg/day benznidazole. At the end of treatment, changes in immune response, cardiac tissue damage, and parasite load were evaluated. The

**Funding:** Agencia Nacional de Investigación y Desarrollo (ANID) BECAS granted IC: 21170501, FG: 21170427 and HQ: 21170968. (URL: http://repositorio.conicyt.cl/hc/es/categories/360001230771-Subdirecci%C3%B3n-de-Proyectos-de-Investigaci%C3%B3n-Fondecyt-). Agencia Nacional de Investigación y Desarrollo (ANID) FONDECYT Granted SF: FONDECYT N° 3210667, UK: FONDECYT N° 1190341, CC: FONDECYT N° 3180452, GD: FONDECYT N° 1210627, and JDM: FONDECYT N° 1210359 (URL: https://ayuda.anid.cl/hc/es/categories/360001230771-Subdirecci%C3%B3n-de-Proyectos-de-Investigaci%C3%B3n-Fondecyt-). Conselho Nacional de Desenvolvimento Científico e Tecnológico (CNPq: 305894/2018-8 for FSM) and Fundação de Amparo a Pesquisa de Minas Gerais (FAPEMIG: Rede Mineira de Imunobiológicos; REDE-00140-16 for FSM), Coordenação de Aperfeiçoamento de Pessoal de Nível Superior (CAPES-Brazil) and National Institute for Science and Technology in Dengue and Host-microbial interactions (APQ-03606-17 for FSM). https://www.gov.br/cnpq/pt-br; http://www.fapemig.br/pt/; and https://www.gov.br/capes/pt-br. The funders had no role in study design, data collection and analysis, decision to publish, or preparation of the manuscript.

**Competing interests:** The authors have declared that no competing interests exist.

administration of AT-RvD1 in the early chronic phase of *T. cruzi* infection regulated the inflammatory response both at the systemic level and in the cardiac tissue, and it reduced cellular infiltrates, cardiomyocyte hypertrophy, fibrosis, and the parasite load in the heart tissue.

## Conclusions/Significance

AT-RvD1 was shown to be an attractive therapeutic due to its regulatory effect on the inflammatory response at the cardiac level and its ability to reduce the parasite load during early chronic *T. cruzi* infection, thereby preventing the chronic cardiac damage induced by the parasite.

## Author summary

Chagas disease is prevalent in Latin America and is widely distributed worldwide due to migration. In 30% of patients, if the parasite is left untreated, the disease may progress from an acute symptomless phase to chronic myocardial inflammation, which can cause heart failure and death, years after the infection. Imbalances in the inflammatory response are related to this progression. Current treatments cannot prevent or reverse the cardiac damage inflicted by the parasite. Aspirin-triggered resolvin D1, also named AT-RvD1, can modify cellular and humoral inflammatory responses leading to the resolution of inflammation, thus promoting healing and restoring organ function. In this study, AT-RvD1, in an N-formyl peptide receptor 2 (FPR2)-dependent manner, was shown to regulate local and systemic inflammation and decrease cellular infiltration in the heart tissue of mice chronically infected with the parasite and reduce cardiac hypertrophy and fibrosis in the early stages of the chronic phase of the disease. Importantly, AT-RvD1 was able to decrease parasite load in the infected hearts. Thus, this research indicates that At-RvD1 treatment is a potential therapeutic strategy that offers an improvement on current drug therapies.

## Introduction

Chagas disease (CD) is caused by the protozoan *Trypanosoma cruzi*, which afflicts 7 million people in 21 endemic Latin American countries and is increasing in non-endemic countries due to migration [1]. According to the World Health Organization [1], the most alarming statistics are that 30 000 new cases, of which 8000 are newborns, are reported annually and there are more than 10 000 deaths per year. Clinically, CD initially presents an acute phase, generally asymptomatic, but in the absence of treatment a chronic phase develops. In 30% of chronic cases, patients develop cardiac or digestive complications 10–30 years after acquiring the infection [2,3]. Chronic Chagas cardiomyopathy (CCC) is considered the most frequent and severe clinical manifestation of CD. It is characterized by focal inflammatory infiltrates, cardiac hypertrophy, and fibrosis, leading to abnormalities in the electrical conduction system, heart failure, and sudden death [4,5]. Chronic inflammation is considered one of the most important mechanisms involved in the pathogenesis of CD and has been proposed as a consequence of tissue damage due to the persistence of live parasites [6]. This chronic inflammation is a determining factor in the deterioration of heart architecture and loss of functionality [7]. The

current treatment for CD utilizes nitroheterocyclic drugs, such as benznidazole (Bz), which are effective mainly in the acute phase and in the treatment of congenital transmission. Recent studies have shown that these drugs can prevent the progression of the chronic phase [8,9]. However, adherence to treatment with these drugs may be compromised by adverse events that can force the discontinuation of the therapy [10,11]. Therefore, new approaches for treating this disease or improving the action of current antiparasitic drugs are needed [12].

Acute inflammation is a natural protective mechanism of the host in response to injury or invading pathogens. The resolution of inflammation is an active process orchestrated by molecules known as specialized pro-resolving mediators (SPM) that facilitate the termination of the inflammatory processes [13]. Endogenous SPMs actively participate in the dampening of host responses and the resolution of inflammation. Mainly produced by macrophages and neutrophils from distinct omega-3 polyunsaturated fatty acid pathways, the four families of SPMs, lipoxins, maresins, protectins, and resolvins, have been shown to act as initiators of the resolution of acute inflammation; therefore, they limit polymorphonuclear neutrophil infiltration, counteract the production of cytokines and chemokines, and enhance macrophage-mediated actions [14]. Resolvin D1 (RvD1) is a novel SPM whose effects on inflammatory diseases dampen pathological inflammatory responses and restore tissue homeostasis [15]. Some drugs such as acetylsalicylic acid (ASA) modify the activity of cyclooxygenase 2 (COX-2), allowing SPM epimer generation, specifically 15-epi-lipoxin A4 (15-epi-LXA$_4$) and AT-RvD1-4 [16,17]. These "aspirin-triggered" SPMs have the advantage of being more stable to enzymatic degradation than endogenous molecules and can serve as anti-inflammatory drugs [16,18].

In CD, ASA has been extensively studied and beneficial effects have been reported closely related to its dose. In experimental models, a low dose of ASA (25 mg/kg) improves survival, reduces heart inflammatory infiltrates, and improves cardiac tissue architecture. These effects were associated with a significant increase in the production of 15-epi-LXA$_4$ [19]. Moreover, this aspirin-triggered lipoxin reduces the internalization of *T. cruzi* in macrophages [20], an effect mediated by N-formyl peptide receptor 2 (FPR2) [21,22]. These effects have highlighted the role of FPR2 in the beneficial effects of ASA. In recent years, the effects of AT-RvD1 on CD have been reported in several publications. First, on peripheral mononuclear cells (PBMCs) from patients with stage B1 Chagas heart disease (having few cardiac abnormalities), AT-RvD1 had an immunomodulatory effect by decreasing the production of pro-inflammatory cytokines such as interferon-gamma (IFNγ) and the proliferation of PBMCs after stimulation with *T. cruzi* antigen, counteracting the inflammatory environment [23]. Second, the effect of RvD1 was recently studied in a murine model chronically infected with *T. cruzi*, where RvD1 therapy increased the survival rate and regulated the inflammatory response by reducing serum IFNγ levels and increasing serum IL-10 levels. Furthermore, RvD1 reduced inflammatory infiltrates, favoring the resolution of *T. cruzi* infection and preventing cardiac fibrosis [24].

After infection in mammals, *T. cruzi* induces robust innate and adaptive immune responses, which play a significant role during the acute and chronic phases of the disease. Nonetheless, these responses are insufficient to achieve complete clearance of the parasite, and parasite persistence promotes low and sustained inflammation over time [6]. Therefore, because of the benefits of AT-RvD1 in reducing pathological inflammatory processes and promoting restoration of tissue homeostasis, it is possible that AT-RvD1 administration in CD aids in the prevention of inflammatory damage secondary to parasite persistence and in parasite clearance. In the present study, we evaluated the effects of AT-RvD1 and whether Bz and AT-RvD1 combined therapy could improve parasite control and reduce inflammation, heart damage, and irregular cardiac electrical activity, limiting the progress of CCC in a murine

model of early chronic CD. Thus, we have identified AT-RvD1 as a new potential therapeutic approach to modulate the pathogenesis of CD.

## Methods

### Ethics statement

This research study was carried out in strict accordance with the Brazilian Guidelines on animal work and the Guide for the Care and Use of Laboratory Animals of the National Institutes of Health (NIH). The Institutional Bioethics Committee of the Faculty of Medicine, University of Chile, approved the supervising protocols (Protocol CBA# 1078 FMUCH, associated with FONDECYT-Chile grant number 1170126).

### Animals

Animal care and handling procedures were in accordance with the guidelines of the local animal ethics committee. Eight to ten-week-old C57BL/6 wild-type (WT) mice were obtained from the Animal Care Facilities of Universidade Federal de Minas Gerais (UFMG, Brazil). The FPR2 knockout mice were bred on a C57BL/6 genetic background under pathogen-free conditions at the Instituto de Ciências Biológicas–UFMG. The animals were randomly distributed and were housed at 2 to 3 animals per box in a controlled environment at constant temperature, under a 12-h day/night cycle, and with food and water available *ad libitum*.

### Parasites and infection protocols

Dm28c (TcI lineage) trypomastigotes from Vero cell cultures ($1 \times 10^5$) were used to intraperitoneally inoculate the C57Bl/6 mice. At the peak of parasitemia, blood was collected and pooled from various donors, and the parasites were counted by direct visualization of a drop of blood under a light microscope. Finally, the trypomastigotes were suspended in sterile saline, and $1 \times 10^3$ trypomastigotes in 100 μL were injected into each experimental animal. After randomization, the animals were divided into five groups (healthy groups) or eight groups (infections and different treatments). *T. cruzi* infection was confirmed from the third day after infection by direct microscopic visualization of circulating trypomastigotes in peripheral blood samples obtained from the tail tip. Subsequently, the parasitemia was monitored every two days via peripheral blood samples until undetectable [19].

### Treatment protocols

Infected C57BL/6 WT and FPR2$^{-/-}$ mice (eight mice per group: four males and four females) were treated with 5 μg/kg/day aspirin-triggered Resolvin D1 (AT-RvD1) (Cayman Chemicals, Ann Harbor, MI, USA) equivalent to 100 ng/mouse AT-RvD1 [16,24], 30 mg/kg/day Bz (Abarax—Elea Laboratory, Buenos Aires, Argentina), or a combination of 5 μg/kg/day AT-RvD1 + 5 mg/kg/day Bz [25]. Bz was suspended in 0.5% carboxymethylcellulose and administered orally by gavage once a day. AT-Rv1 was dissolved in 0.01% ethanol as vehicle and administered intraperitoneally once a day. Untreated control mice (three males and two females) received vehicle intraperitoneally. The treatments were administered from day 40 to 60 post-infection (p.i.), simultaneously. On day 60 p.i., the animals were anesthetized with 100 mg/kg ketamine and 10 mg/kg xylazine and their blood and hearts were removed. The left ventricles were isolated and halved, and one half was fixed in 4% formaldehyde in 0.1 M phosphate-buffered saline (PBS; pH 7.3) for histological analysis and the other half was homogenized using a Heidolf homogenizer (Heidolf RZR 2050, Heidolph Instruments GmbH

& Co. KG). TRIzol (Invitrogen, Waltham, MA, USA) was used to extract mRNA for the analyses of cytokine gene expression and parasite load by RT-qPCR.

## Enzyme-linked immunosorbent assay (ELISA)

Peripheral blood was collected from the submandibular vein of the C57BL/6 mice on days 20 and 40 p.i. On day 60, whole blood was collected after euthanasia. The blood was centrifuged, and the resulting serum was stored at -80˚C. Cytokine quantification was performed on the serum samples by ELISA using specific mouse monoclonal antibodies from the ELISA Max Deluxe Set for mouse tumor necrosis factor alpha (TNFα), IFNγ, interleukin (IL)-1β, and IL-10 (BioLegend, San Diego, CA, USA), following the manufacturer's instructions. The absorbance of the ELISA plate wells was read at 450 nm in a microplate reader Biotek (Winooski, VT, USA). All samples were analyzed in duplicate.

## Quantitative reverse transcription PCR (RT-qPCR) assay

For measurements of cytokine gene expression in cardiac tissue, total RNA was isolated using TRIzol reagent, followed by DNAse treatment and purification using the PureLink RNA Mini Kit (Thermo Fisher Waltham, MA, USA) according to the manufacturer's instructions. cDNA was synthesized from 600 ng of total RNA by reverse transcription using M-MLV reverse transcriptase and random primers (Invitrogen). For the qPCR analyses, each reaction mix contained 150 nM of each primer (forward and reverse), 1 ng of sample cDNA, 7 μL of SensiMix SYBR Green Master Mix (Bioline, Memphis, TE, USA), and $H_2O$ in a total volume of 15 μL. The primers used for the analysis of cytokines and hypertrophy markers were the following: TNFα, Fw: 5′-TAGCCCACGTCGTAGCAAAC-3′ and Rv: 5′-ACAAGGTACAACCCATC GGC-3′; IFNγ, Fw: 5′-AACTGGCAAAAGGATGGTGAC-3′ and Rv: 5′-TTGCTGATGGC CTGATTGTC-3′; IL-1β, Fw: 5′-TGCCACCTTTTGACAGTGATG-3′ and Rv: 5′-GTGCT GCTGCGAGATTTGAA-3′; IL-10, Fw: 5′-ACCTGGTAGAAGTGATGCCC-3′ and Rv: 5′-AC AGGGGAGAAATCGATGACAG-3′; atrial natriuretic peptide (ANP), Fw: 5′-GGGCTTCTT CCTCGTCTTGG-3′ and Rv: 5′-GTGGTCTAGCAGGTTCTTGAAAT-3′; brain natriuretic peptide (BNP), Fw: 5′-CAGAGCAATTCAAGATGCAGAAGC-3′ and Rv: 5′-CTGCCTTGA-GACCGAAGGAC-3′. The amplification was performed in an ABI Prism 7300 sequence detector (Applied Biosystems, Waltham, MA, USA). The cycling program was as follows: a denaturation step at 95˚C for 3 min and 40 amplification cycles of 95˚C (15 s), 60˚C (15 s), and 72˚C (30 s). The final step was a dissociation stage that ramped from 60 to 95˚C over 100 s. For relative quantification, the results were expressed as RQ values determined using the comparative control (delta-delta-Ct (DDCt)) method [26].

The presence of viable parasites in cardiac tissue was evaluated by amplification of 18S *T. cruzi* ribosomal RNA (rRNA). For this purpose, total RNA was isolated using TRIzol and purified using the PureLink RNA Mini Kit; cDNA was synthesized as detailed above. RT-qPCR was performed as described above with the following primers: 18S, Fw: 5′-TGGAGATTATG GGGCAGT-3′ and Rv: 5′-GTTCGTCTTGGTGCGGTCTA-3′. The parasitic load of *T. cruzi* in the cardiac tissues was calculated from a standard curve constructed using $1 \times 10^8$ trypomastigotes of *T. cruzi*, serially diluted to provide a log curve in the range of 1 to $10^8$ equivalent parasites/10 ng of tissue RNA.

## Histology

The hearts from the euthanized mice were fixed in 4% formaldehyde (pH 7.3). Then, the fixed heart tissues were dehydrated with 50 to 100% ethanol, clarified with xylol, embedded in paraffin, and sectioned in 5 μm slices. The sections were stained with hematoxylin and eosin to

observe cellular infiltration and cardiomyocyte cross-sectional area or with picrosirius red to observe collagen organization. Images were obtained using a spinning-disk microscope (Olympus BX42). Five fields per heart (40X) were analyzed using Image J software.

For the analysis of cellular infiltrates, the nuclei present in five fields of each heart were quantified. The number of nuclei per tissue area was counted in order to eliminate empty tissue areas. In the analysis of cardiomyocyte cross-sectional area, only muscle fibers with well-defined borders and a nucleus inside were included. For the analysis of fibrosis in the picrosirius red-stained slides, the red-colored pixels in the cardiac tissue were quantified in five fields per animal.

## Electrocardiogram (ECG) recording and analysis

ECG recording was performed using a six-channel non-invasive electrocardiograph (ECG-PC version 2.07, Electronic Technology of Brazil (TEB), Belo Horizonte/MG, Brazil). The mice were anesthetized initially with 2.5% isoflurane and then maintained with 1.5% isoflurane (VetCase-Incotec, Serra/ES, Brazil). The mice were placed in a dorsal recumbent position on a wooden table covered with plastic material; electrocardiographic gel was applied, and four alligator clip electrodes were attached to the skin of the forelimbs and hindlimbs. All procedures were performed in a quiet room to minimize stress.

All ECGs were performed and analyzed by the same technician according to standard methods for ECG trace analysis. The tracings were recorded from six leads of the frontal plane at a velocity of 50 mm/s. In each tracing, three segments containing five beats (lead II) were selected for quality (clean baseline with no artifacts), and the mean values for heart rate (HR) and duration of the intervals and waves were determined. The parameters evaluated were heart rate, P wave, QRS complex, PR interval, and QT interval. QT-corrected values were obtained from Bazett's formula.

## Statistical analysis

For all experiments, statistical significance was established at p values of 0.05. The data represent the means ± standard deviations (SD) from at least three independent observations or experiments. All statistical analyses were performed using GraphPad Prism 8.0 software. One-way and two-way analyses of variance (ANOVAs) (with Tukey's post-hoc tests) were performed as appropriate. A log-rank test was performed for survival analysis.

# Results

## Effect of the absence of FPR2 on *T. cruzi* parasitemia

The progression of parasitemia and mortality in C57BL/6 WT and FPR2$^{-/-}$ mice infected with *T. cruzi* strain Dm28c were determined to evaluate the establishment of a murine model for chronic CD (Fig 1). As expected, in the WT mice, detectable parasitemia persisted for approximately 40 days, with the peak of infection occurring at 27 days p.i. (Fig 1A). The features of acute infection in both mice groups were very similar, although a difference was observed between the two groups on day 27 of parasitemia.

Furthermore, survival rates were high for all mice until the end of the experiment, regardless of their genetic background, confirming that the mortality of this murine model of *T. cruzi* chronic infection was low.

Regarding inflammatory responses, changes in serum levels of pro-inflammatory (TNFα, IFNγ, IL-1β) and anti-inflammatory cytokines (IL-10) were determined at days 20, 40, and 60 p.i. in WT and FPR2$^{-/-}$ mice infected with *T. cruzi* (S1A, S1C, S1E and S1G Fig). Both the

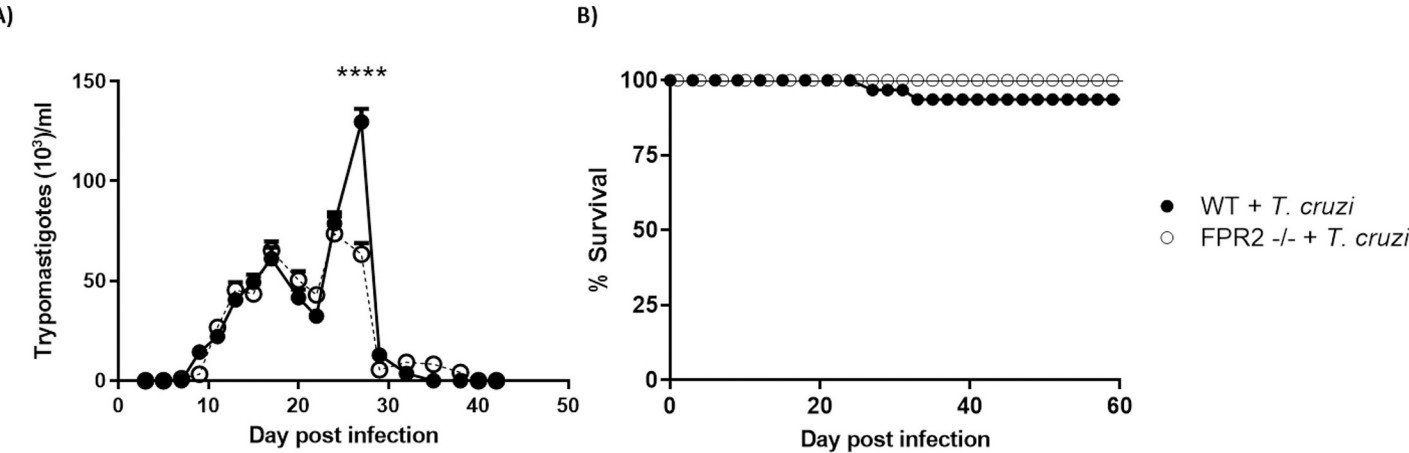

**Fig 1. Parasitemia and survival in C57BL/6 WT and FPR2 -/- mice infected with *Trypanosoma cruzi*.** A) Blood parasite levels, B) Kaplan-Meier survival curves. Data are expressed as the mean ± SD (*n* = 32 mice per group). Two-way ANOVA was performed to identify significant differences, \*\*\*\*p ≤ 0.0001. WT, Wild-type; FPR2 -/-, FPR2 knock-out

healthy WT and FRP2[-/-] mice showed differences in serum cytokine levels at baseline and after *T. cruzi* infection. The healthy FPR2[-/-] mice had elevated levels of TNFα and IL-10 compared to those of the healthy WTs (S1A and S1G Fig). In addition, the magnitude of the inflammatory response was higher in the WT mice than in the FPR2[-/-] mice (S1B, S1D and S1H Fig). At day 40 p.i., the infected WT mice still maintained slightly increased TNFα, IFNγ, and IL-10 levels compared to their healthy controls, while the FPR2[-/-] mice only had slightly increased IFNγ levels. These findings indicate that the FPR2[-/-] mice developed a different immune response against *T. cruzi* infection than the WT mice (S1 Fig).

### Effect of AT-RvD1 treatment on the inflammatory state in CD

The effect of AT-RvD1 treatment on inflammation in early chronic CD was evaluated by measuring serum cytokine levels at the end of treatment (60 dpi). Consistent with a chronic inflammatory state, the serum levels of pro-inflammatory cytokines such as TNFα (Fig 2A), IFNγ (Fig 2B), and IL-1β (Fig 2C) increased as a result of the infection in the WT and FPR2[-/-] mice, although the IL-1β increase in the FPR2[-/-] mice was not significant. Unexpectedly, AT-RvD1 did not reduce serum TNFα levels in the WT mice (Fig 2A); however, AT-RvD1 treatment significantly reduced the serum levels of IFNγ (Fig 2B) and IL-1β (Fig 2C), reaching healthy control levels. This effect was mediated by FPR2 because in the FPR2[-/-] mice, AT-RvD1 treatment did not alter IFNγ levels. In contrast, Bz, alone or combined with AT-RvD1, did not affect the increased serum levels of pro-inflammatory cytokines, except for IL-1β in the WT mice (Fig 2C). Both treatments produced negligible effects on Il-10 (Fig 2D).

Because serum cytokine levels reflect systemic inflammatory states, it was also necessary to measure them in the heart to determine local inflammatory states. Thus, the effect on the immune response in cardiac tissue was determined by measuring relative cytokine mRNA levels after 20 days of treatment with AT-RvD1 (60 dpi). In this case, mRNA levels of the pro-inflammatory TNFα (Fig 3A), IFNγ and IL-1β (Fig 3C) cytokines were increased in both the WT and FPR2[-/-] infected mice. Although the mRNA levels of IFNγ increased exceedingly upon infection, it was not modified by At-RvD1 or Bz in WT and FPR2[-/-] mice (Fig 3B). Interestingly, in the WT and FPR2[-/-] mice, AT-RvD1 treatment significantly increased IL-10 expression levels (Fig 3D). AT-RvD1 prevented the increase in TNFα caused by infection. Neither Bz nor combinatorial therapy modified the mRNA levels of the measured cytokines.

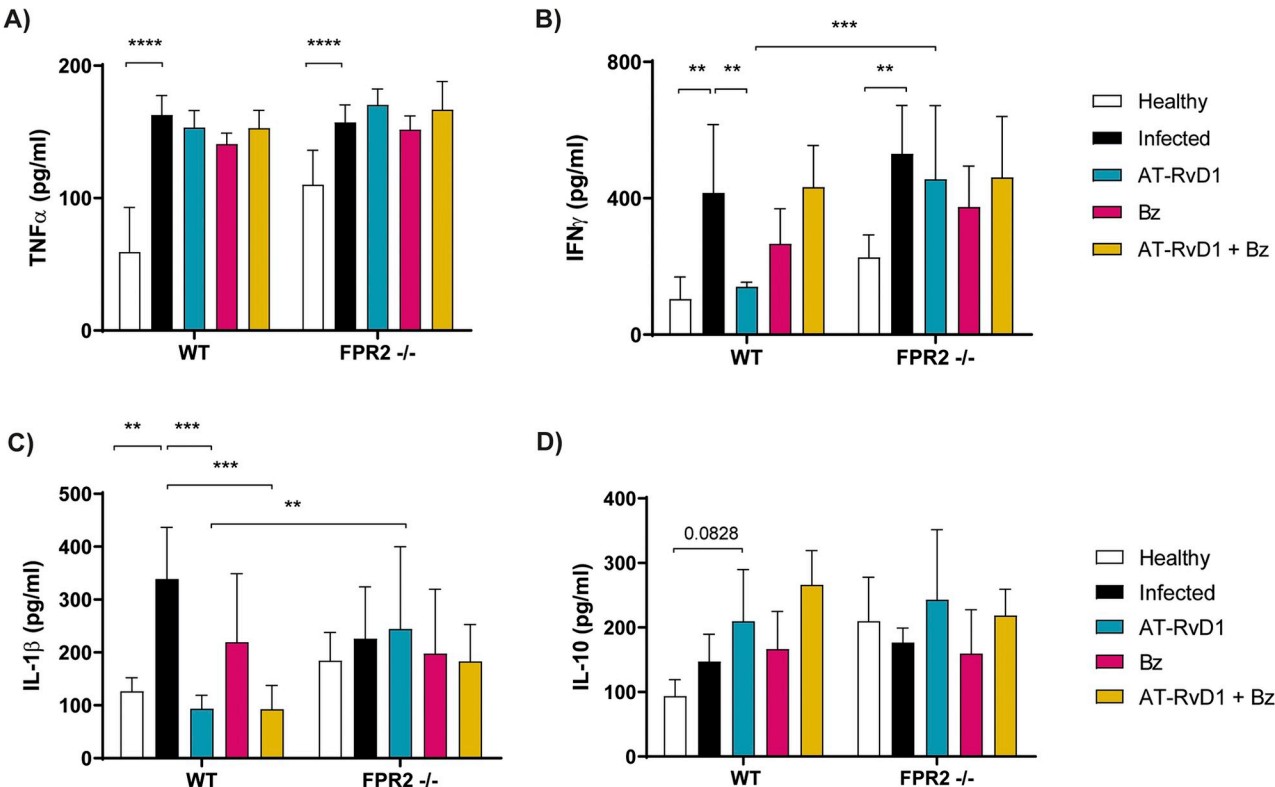

**Fig 2. Serum cytokine levels in mice infected with *Trypanosoma cruzi* during early chronic infection.** Mice were infected with *T. cruzi* (Dm28c) trypomastigotes and treated with 5 μg/kg/day AT-RvD1 or 30 mg/kg/day Bz between day 40 and 60 post-infection. For the combination treatment, 5 μg/kg/day AT-RvD1 and 5 mg/kg/day Bz were used. At the end of treatment, serum was collected for the measurement of cytokine concentration. The concentrations of TNFα (A), IFNγ (B), IL-1β (C), and IL-10 (D) in serum were determined by ELISA. Data are expressed as the mean ± SD (*n* = 8 mice per group). Two-way ANOVAs and Tukey's post-hoc tests were performed to identify significant differences. $^{**}p \leq 0.01$, $^{***}p \leq 0.001$, $^{****}p \leq 0.0001$. Bz, benznidazole; AT-RvD1, aspirin-triggered resolvin D1.

In CCC, chronic inflammation is a determining factor in the deterioration of heart architecture. The cellular infiltrates in heart tissue sections were measured to evaluate the effect of AT-RvD1 treatment on cardiac inflammation. Histological analysis of cardiac tissue stained with hematoxylin-eosin was performed, and cellularity was quantified (Fig 4A). WT and FPR2$^{-/-}$ mice infected with *T. cruzi* (60 dpi) showed focal inflammatory infiltrates and increased cellularity in cardiac tissue. AT-RvD1 significantly reduced focal inflammatory infiltrates in the cardiac tissue (Fig 4B); the absence of FPR2 impaired the reduction in cellular infiltrates, suggesting that this receptor mediates the AT-RvD1 effect. Moreover, the effect of AT-RvD1 in reducing the inflammatory infiltrates had a higher significance level than that obtained with Bz alone. Consistently, the combination of AT-Rv1 with Bz significantly reduced the inflammatory infiltrates in the WT mice, whereas, in the absence of FPR2, the reduction in inflammatory infiltrates achieved by combinatorial therapy was probably dependent on the effect of Bz.

## Through FPR2, AT-RvD1 prevents cardiac remodeling in early chronic CD

Because cardiac hypertrophy and dysfunction are significant complications of CCC, the effect of AT-RvD1 on cardiac hypertrophy was assessed by measuring the cross-sectional area of cardiomyocytes from heart tissue sections stained with hematoxylin-eosin (Fig 5A). *T. cruzi* infection increased the cross-sectional area of cardiomyocytes in the WT and FPR2$^{-/-}$ mice after 60

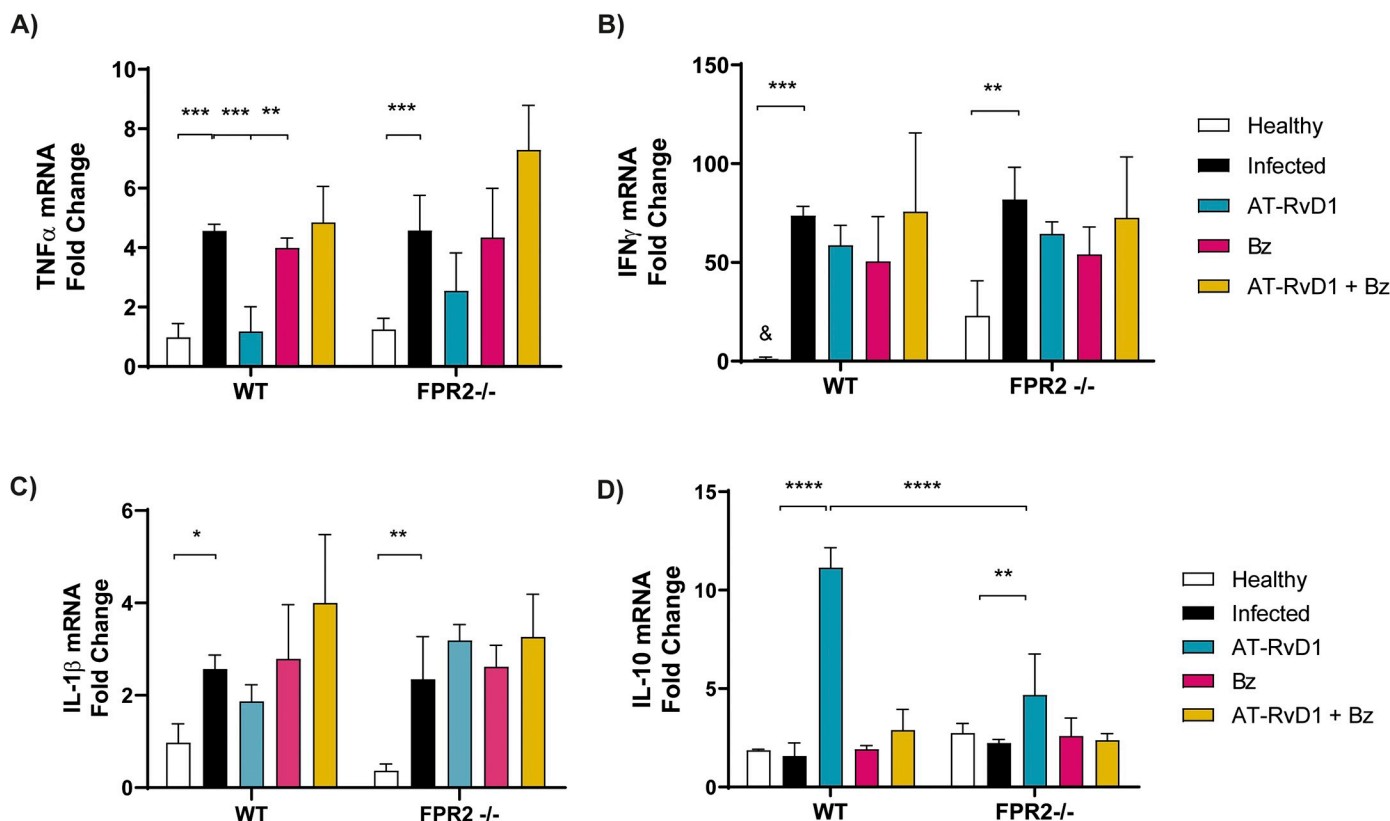

**Fig 3. Cytokine mRNA levels in mice infected with *Trypanosoma cruzi* during early chronic infection.** Mice were infected with *T. cruzi* (Dm28c) trypomastigotes and treated with 5 μg/kg/day AT-RvD1 or 30 mg/kg/day Bz between day 40 and 60 post-infection. For the combination treatment, 5 μg/kg/day AT-RvD1 and 5 mg/kg/day Bz were used. At the end of treatment, mRNA was extracted from the left ventricles. The cardiac mRNA levels of TNFα (A), IFNγ (B), IL-1β (C), and IL-10 (D) are shown from uninfected C57BL/6 mice infected with *T. cruzi* and treated with AT-RvD1 and Bz, as assessed by RT-qPCR. Data are expressed as the mean ± SD (*n* = 8 mice per group). Two-way ANOVAs and Tukey's post-hoc tests were performed to identify significant differences. *p ≤ 0.05, **p ≤ 0.01, ***p ≤ 0.001, ****p ≤ 0.0001. & indicates indeterminate values. Bz, benznidazole; AT-RvD1, aspirin-triggered resolvin D1.

dpi, demonstrating structural changes in the earliest phases of chronic disease. It is striking that in healthy FPR2$^{-/-}$ mice, the cross-sectional areas were also increased compared with those in the WT mice, which was consistent with the overall increase in circulating cytokine levels (Fig 2). Although AT-RvD1 significantly reduced cardiomyocyte cross-sectional area, the effect was blunted by the absence of FPR2 because in the FPR2$^{-/-}$ mice, there was no reduction in cardiac hypertrophy (Fig 5B). It is important to note that Bz alone did not affect cardiac hypertrophy; however, when combined with AT-RvD1, it significantly reduced the cross-sectional area of cardiomyocytes in the WT mice, but not in the absence of FPR2. These results suggest that AT-RvD1 therapy can reduce cardiac hypertrophy. These observations were supported by measuring the mRNA levels of ANP and BNP, molecular markers of cardiac hypertrophy (Fig 5C and 5D, respectively). Both ANP and BNP were significantly increased in the infected WT and FPR2$^{-/-}$ mice (60 dpi), although the changes in the FPR2$^{-/-}$ mice were negligible. Consistent with the histological analyses, treatment with AT-RvD1 significantly reduced both molecular markers of cardiac hypertrophy. However, the absence of FPR2 did not change the effect of AT-RvD1, suggesting that an alternative way mediated this effect. Bz alone or combined with AT-RvD1 did not modify these molecular markers of hypertrophy.

Although cardiac hypertrophy in chronic CD is a hallmark of cardiac remodeling, cardiac fibrosis reflects more profound structural damage. Thus, the effect of AT-RvD1 on cardiac

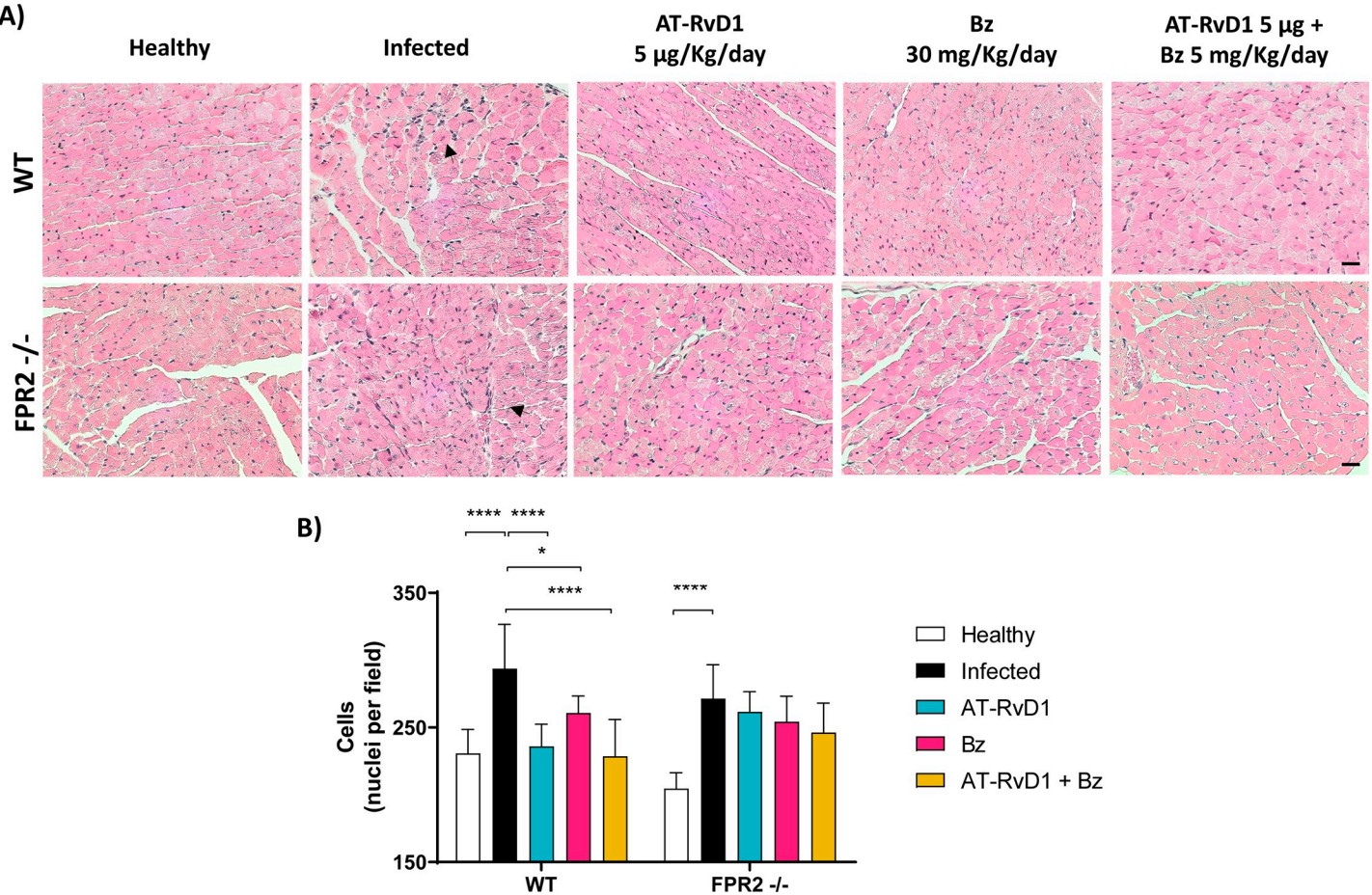

**Fig 4. AT-RvD1 reduces cellular infiltrates in the heart tissue of C57BL/6 mice infected with *Trypanosoma cruzi* for 60 days.** Mice were infected with *T. cruzi* (Dm28c) trypomastigotes and treated with 5 µg/kg/day AT-RvD1 or 30 mg/kg/day Bz between day 40 and 60 post-infection. For the combination treatment, 5 µg/kg/day AT-RvD1 and 5 mg/kg/day Bz were used. At the end of treatment, the hearts were removed, and the left ventricles were analyzed. A) Representative images of the left ventricles stained with hematoxylin and eosin from C57BL/6 mice infected with *T. cruzi*. Black arrows show infiltrated cells. Scale bar = 20 µm. B) Quantitative analysis of cellularity in the ventricular tissues from five fields per animal. Data are expressed as the mean ± SD (*n* = 8 mice per group). Two-way ANOVA and Tukey's post-hoc tests were performed to identify significant differences. *p ≤ 0.05, ****p ≤ 0.0001. AT-RvD1, aspirin-triggered resolvin D1.

fibrosis in CCC was analyzed in cardiac tissue stained with picrosirius red to visualize collagen fibers (Fig 6A). Consequently, cardiac fibrosis, expressed as collagen deposition and alpha-smooth muscle actin (α-SMA) expression (Fig 6B and 6C) was increased with CD progression in both WT and FPR2[-/-] infected mice (60 dpi), and AT-RvD1 significantly reduced this cardiac fibrosis, more efficiently than Bz in both genotypes.

Signs of cardiac remodeling include atrioventricular and intraventricular conduction disorders such as right bundle branch block, sinus bradycardia, and QT interval changes, alterations that may occur early p.i. [27,28]. Thus, the electrocardiographic activities of the C57BL/6 WT and FPR2[-/-] mice were evaluated at the end of the treatments. However, at 60 dpi, no significant changes were found in the ECG parameters of the mice (S2 Fig).

## AT-RvD1 reduces parasite load in early chronic CD

Cardiac parasite load was determined to verify establishment of the chronic phase and impact of the treatments on parasite clearance. For this determination, RT-qPCR of *T. cruzi* 18S rRNA was performed to determine parasite numbers in the cardiac tissue at the end of the

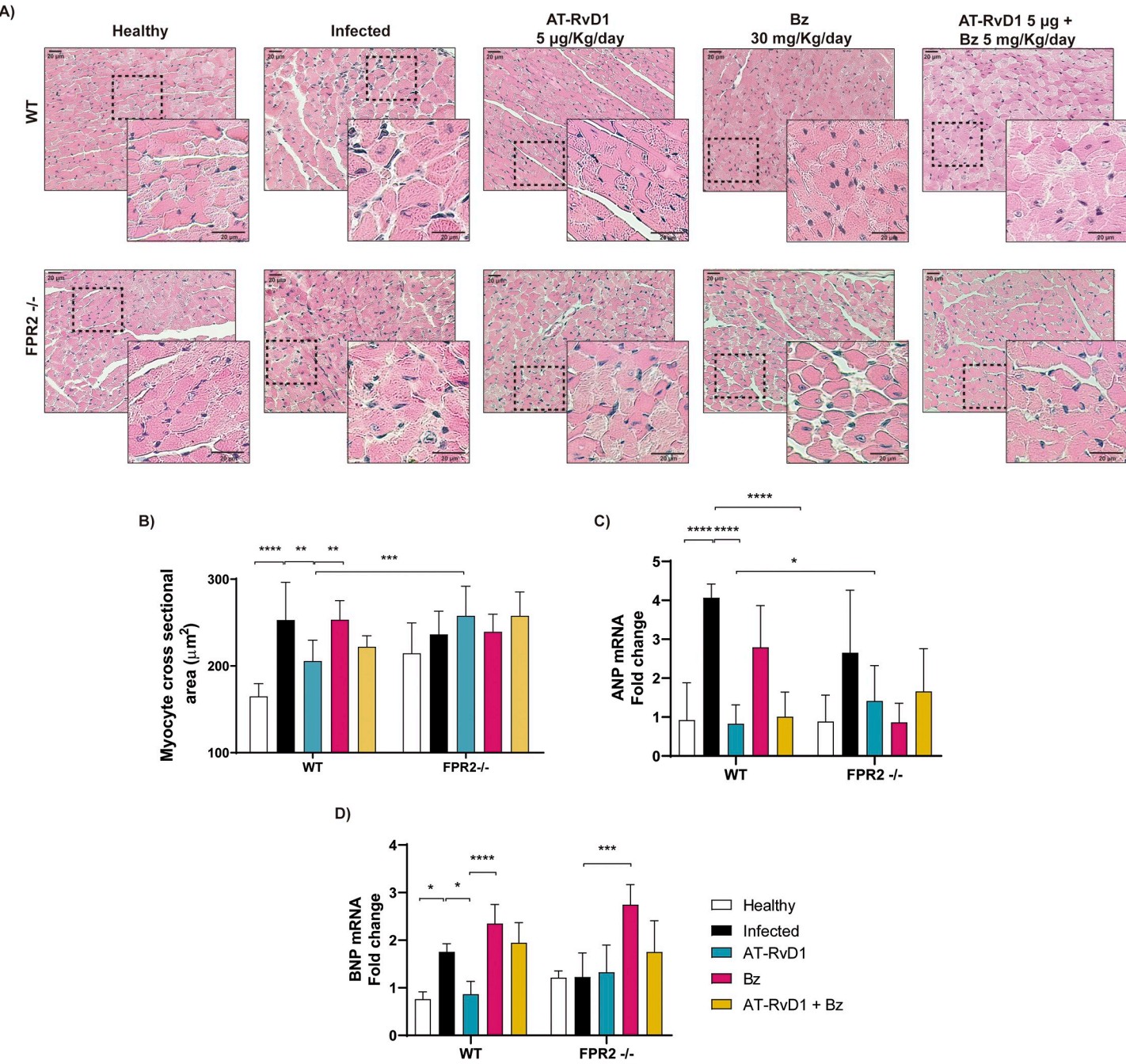

**Fig 5. AT-RvD1 reduces cardiac hypertrophy in C57BL/6 mice infected with *Trypanosoma cruzi* for 60 days.** Mice were infected with *T. cruzi* (Dm28c) trypomastigotes and treated with 5 μg/kg/day AT-RvD1 or 30 mg/kg/day Bz between day 40 and 60 post-infection. For the combination treatment, 5 μg/kg/day AT-RvD1 and 5 mg/kg/day Bz were used. At the end of treatment, the hearts were removed and the left ventricles were analyzed. A) Images are enlargements of the representative images shown in Fig 4. B) Analysis of the cross-sectional area of cardiomyocytes; 200 cardiomyocytes were randomly chosen per animal. Scale bar = 20 μm. C, D) mRNA levels of atrial natriuretic peptide (ANP) and brain natriuretic peptide (BNP), as markers of hypertrophy, are shown from cardiac tissue of C57BL/6 mice infected with *T. cruzi* and treated with AT-RvD1 and/or Bz, as measured by RT-qPCR. Data are expressed as the mean ± SD (*n* = 8 mice pre group). Two-way ANOVAs and Tukey's post-hos tests were performed to identify significant differences. $^*$p ≤ 0.05, $^{**}$p ≤ 0.01, $^{***}$p ≤ 0.001, $^{****}$p ≤ 0.0001. Bz, benznidazole; AT-RvD1, aspirin-triggered resolvin D1.

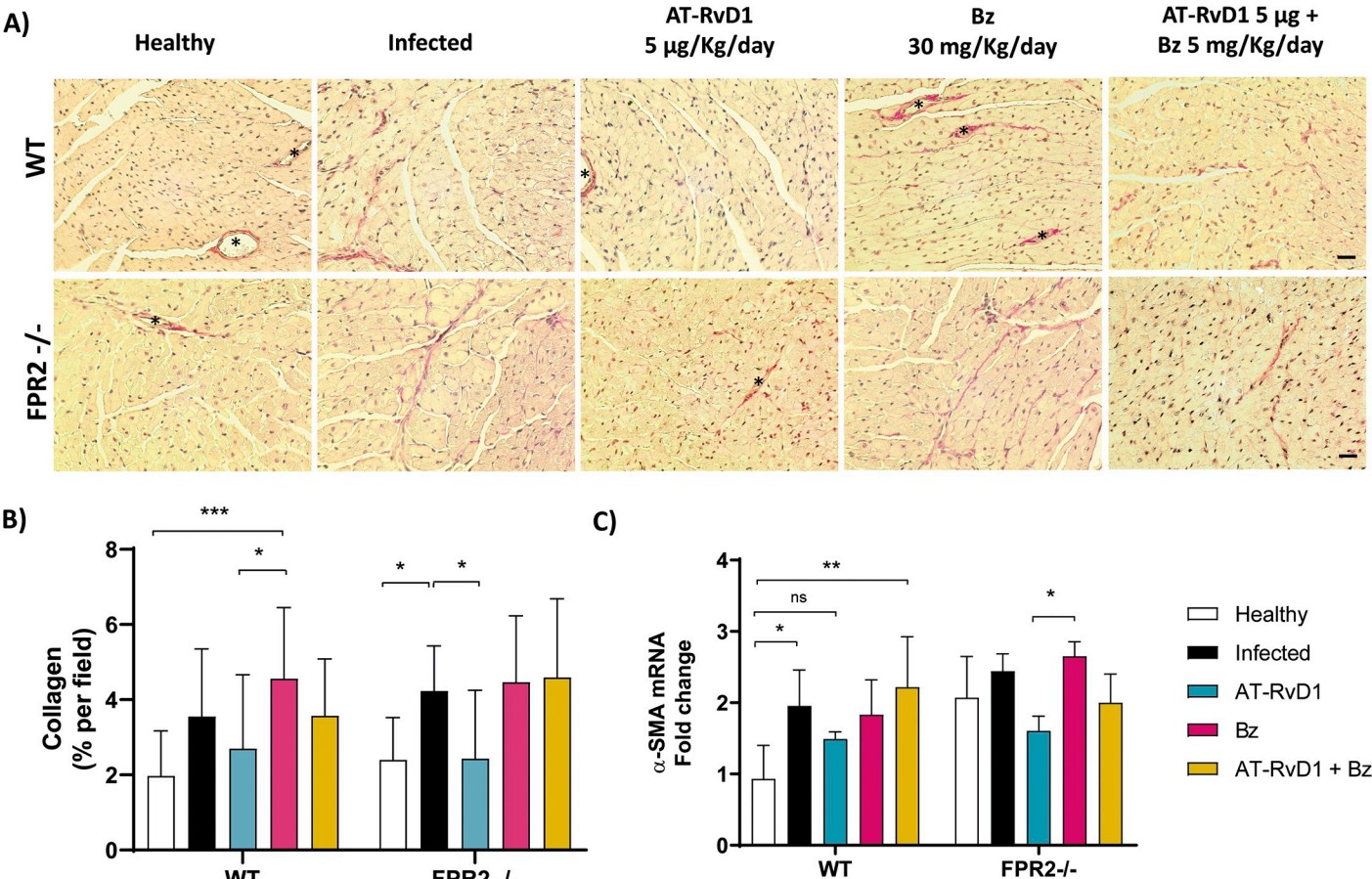

**Fig 6. AT-RvD1 reduces cardiac fibrosis in C57BL/6 mice infected with *Trypanosoma cruzi* for 60 days.** Mice were infected with *T. cruzi* (Dm28) trypomastigotes and treated with 5 μg/kg/day AT-RvD1 or 30 mg/kg/day Bz between day 40 and 60 post-infection. For the combination treatment, 5 μg/kg/day AT-RvD1 and 5 mg/kg/day Bz were used. At the end of treatment, the hearts were removed and the left ventricles were analyzed. A) Representative images of picrosirius red-stained left ventricular tissue from C57BL/6 mice infected with *T. cruzi*. Asterisks show blood vessels. Scale bar = 20 μm. B) Quantitative analysis of red pixels in the stained left ventricular tissue, corresponding to regions of fibrosis, from five fields per animal. C) Relative change in α-SMA expression in the left ventricles from C57BL/6 mice infected with *T. cruzi*. Data are expressed as the mean ± SD ($n$ = 8 mice per group) A two-way ANOVA and Tukey's post-hoc tests were performed to identify significant differences. $^*$p $\leq$ 0.05, $^{***}$p $\leq$ 0.001. AT-RvD1, aspirin-triggered resolvin D1.

treatments. As expected, Bz reduced cardiac *T. cruzi* load to undetectable levels in both the WT and FPR2$^{-/-}$ mice (Fig 7). However, a surprising finding was that AT-RvD1 decreased the cardiac *T. cruzi* load in the WT and FPR2$^{-/-}$ mice, although the decrease in parasitic load was less marked in the null mice. This finding was unexpected because AT-RvD1 does not have trypanocidal properties.

## Discussion

### Involvement of inflammation resolution in parasitemia control

In the experimental model used, the course of parasitemia in the WT and FPR2$^{-/-}$ mice was representative of the behavior expected by the Dm28c strain of *T. cruzi* [19,29]. Although there was no difference between the characteristics of parasitemia observed in the two mice groups in the present study, other study reported significantly lower parasitemia in the same FPR2-/- mice, than that observed in WT mice [30]. The latter results are consistent with other experimental models, wherein the deletion of the gene encoding 5-lipoxygenase (5-LO), which

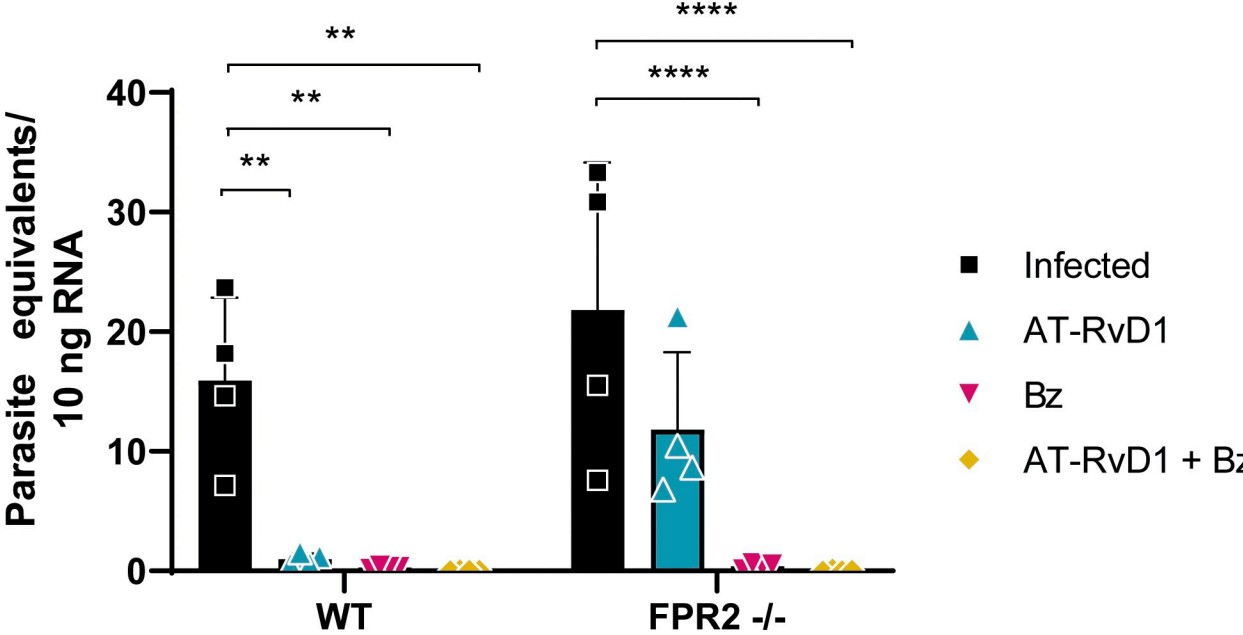

**Fig 7. Parasite load in cardiac tissue of mice infected with *Trypanosoma cruzi* for 60 days.** Mice were infected with *T. cruzi* (Dm28c) trypomastigotes and treated with 5 μg/kg/day AT-RvD1 or 30 mg/kg/day Bz between day 40 and 60 post-infection. For the combination treatment, 5 μg/kg/day AT-RvD1 and 5 mg/kg/day Bz were used. At the end of treatment, the hearts were removed and the left ventricles were homogenized for mRNA extraction. RT-qPCR was used to detect 18S rRNA from *T. cruzi*. Data are expressed as the mean ± SD (*n* = 8 mice per group). Two-way ANOVA and Tukey's post-hoc tests were performed to identify significant differences. **p ≤ 0.01, ****p ≤ 0.0001. Bz, benznidazole; AT-RvD1, aspirin-triggered resolvin D1.

participates in the biosynthesis of lipoxins, also showed reduced parasitemia [25,30]. However, in other studies, 5-LO deficiency led to increased parasitemia without affecting survival or infection control [31,32]. These results suggest that the breakdown of one or more pro-resolving elements of inflammation allows better control of the parasite during the acute phase of the disease. This is probably caused by less containment of the inflammatory response and an increased macrophage capability to eliminate parasites. Specifically, parasitemia in the FPR2[-/-] mice was reduced after three weeks p.i., the time when an adaptive antigen-specific response is established and cytotoxic and helper T lymphocytes and B cells participate [33]. These responses could be more efficient in the FPR2[-/-] mice.

The WT and FPR2[-/-] mouse survival was similar and elevated throughout the experimental timeline, probably because the C57BL/6 mouse lineage is resistant to infection with *T. cruzi* [34]. Interestingly, healthy FPR2[-/-] mice had elevated basal levels of IL-10 (S1 Fig), which could be related to a lower risk of mortality during the acute phase of infection. In contrast, infected IL-10 knockout mice have a poor survival rate [35]. Thus, the anti-inflammatory balance mediated by IL-10 likely plays a fundamental role in the survival and control of deleterious inflammatory responses, probably mediated by Treg cell functions [36,37].

The transition from acute to chronic phase is accompanied by decreased tissue parasitism and blood parasitemia and control of the inflammatory response [38], which was observed in our experimental model at 40 p.i.

The virulence of the strain involved in the pathogenesis of Chagas disease is a crucial determinant of the progression of its chronic phase. Therefore, it is possible to find different strains in a patient due to different kinds of exposure. Furthermore, the distribution of strains and lineages varies regionally. Therefore, it would be unreasonable to extrapolate results obtained

from a single strain to establish disease pathogenesis. However, it is logical to assume that strains in the TcI lineage may have similar behaviors. Thus, further studies are required to validate this hypothesis. On the other hand, although strain Dm28 has low virulence, the results pertaining to parasitemia and cardiac involvement obtained in our study confirm an effective establishment of infection, and our results point toward further investigation of the possibility of modifying host immunological factors to produce a safer antiparasitic treatment profile.

## AT-RvD1 decreases the inflammatory process in early chronic CD

Our results showed that infection with *T. cruzi* triggers a robust inflammatory response in the acute phase, with the production of inflammatory cytokines, such as IFNγ and TNFα, which have been shown to activate macrophages to eliminate the parasite [38]. At day 60 p.i., the WT mice showed slightly increased but sustained levels of the pro-inflammatory cytokines TNFα, IFNγ, and IL-1β, reflecting an inflammatory state in the early stages of chronicity. Treatment with AT-RvD1 during the early chronic phase significantly reduced the serum levels of IFNγ and IL-1β, producing an immunoregulatory effect of the inflammatory state at the systemic level. These results are consistent with previous studies, which have shown that PBMCs obtained from chagasic patients in stage B1 (the initial stage of CCC) treated with AT-RvD1 had lower IFNγ production than after antigenic re-exposure to *T. cruzi*, and their cell proliferation was reduced, highlighting the immunomodulatory effect of AT-RvD1 [23].

At the cardiac level, the scenario against *T. cruzi* is not very different. An imbalance is found, with high levels of pro-inflammatory cytokines (TNFα, IFNγ, and IL-1β) and practically unchanged levels of anti-inflammatory cytokines (IL-10). This could reflect the cell populations (including macrophages, cytotoxic T lymphocytes, and helper T lymphocytes) infiltrating the cardiac tissue to coordinate parasite control and represents an exacerbated production of cytokines related to tissue destruction [6]. However, the participation of other cardiac cells such as fibroblasts, endothelial cells, and cardiomyocytes has not been excluded [39]. Moreover, treatment with AT-RvD1 generates an immunomodulatory effect by drastically reducing the production of TNFα at the cardiac level and suddenly increasing the production of IL-10. This suggests that AT-RvD1 increases the proportion of IL-10-producing cells (macrophage subpopulations, Treg and Breg cells) in cardiac tissue to regulate the inflammatory environment.

Furthermore, the decrease in TNFα production may be secondary to the increase in IL-10 because the latter is a potent inhibitor of TNFα mRNA expression through activation of the STAT3 transcription factor pathway in human monocytes and macrophages [40]. That would also explain why, at the systemic level, we did not observe a reducing effect of TNFα. High levels of IL-10 have been associated with evasion of the immune response by different pathogens, including *T. cruzi* [41], contributing to increased mortality or persistence of damage [42]. However, high IL-10 levels can also decrease inflammation and pathology; consequently, a pro/anti-inflammatory balance is necessary for the natural evolution of the disease [43]. Nevertheless, elevated levels of IL-10 seem to be related to maintaining adequate cardiac function in indeterminate patients with CD [44–46]. Therefore, the AT-RvD1 effect on IL-10 production in the heart appears to be beneficial for preventing CCC progression since changes in IL-10 levels have been associated with losing the ability to control the inflammatory immune response.

Although IFNγ is produced in the heart tissue by Th1 cell infiltrates in CCC, there was no change in the mRNA of this cytokine after AT-RvD1 treatment. This finding does not correlate with the corresponding IFNγ serum levels. However, AT-RvD1 reduced the inflammatory infiltrates in cardiac tissue and was more efficient than Bz alone, probably mediated by

reduced cellular and vascular adhesion molecules (VCAM, ICAM) [47–49] in conjunction with modulation of the inflammatory environment, as discussed above.

Previous studies have shown that the anti-inflammatory and pro-resolving properties of AT-RvD1 are primarily mediated by FPR2 [49]. In our model, FPR2 participates in regulating pro-inflammatory and anti-inflammatory cytokines (IFNγ and IL-1B) and reducing cellular infiltrates because its absence weakens the effect produced by AT-RvD1. Moreover, At-RvD1 activates G protein-coupled receptor 32 (GPR32) [50]; thus, GPR32 participation may be an alternative pathway for the action of AT-RvD1. Although the mechanism of action of RvD1 is not fully understood, it has been shown to exert its anti-inflammatory effects by inhibiting the NF-κB pathway [51,52], suppressing cytosolic calcium, and decreasing activation of the calcium-sensitive kinase calcium-calmodulin-dependent protein kinase II (CaMKII) [53]. Alternatively, it has been suggested that resolvins of the D series and maresins inactivate GSK3β, increasing the levels of the active IL-10 transcriptional enhancer CREB and decreasing the levels of the NF-κB (p65) signal [54].

Previous studies have shown that Bz, in addition to its trypanocidal properties, reduces cardiac cell infiltrates and the production of inflammatory mediators via inhibition of the NF-κB pathway [55–58], apparently, an effect dependent on IL-10 [59]. However, in the early chronic CD model presented herein, Bz did not regulate production of the pro-inflammatory (TNFα, IFNγ, and IL-1β) and anti-inflammatory (IL-10) cytokines studied. Thus, there was only a slight reduction in the inflammatory infiltrates with Bz treatment. Moreover, the combination of 5 mg/kg Bz and 5 μg/kg AT-RvD1 reduced cardiac inflammatory infiltrates but did not modulate the pro- and anti-inflammatory cytokines studied. Unfortunately, no synergistic or additive effect could be observed between the two drugs. Although both drugs act by inhibiting the NF-κB pathway, Bz could also increase reactive oxygen species production at the tissue level [60,61], contributing to the production and establishment of a pro-inflammatory environment. This possibly explains why, although Bz is effective as a trypanocidal therapy, it does not reduce the cardiovascular events observed in patients with CCC [62]. Moreover, the results suggest that the combination of AT-RvD1 with Bz does not provide a synergistic effect, and in some cases, it could even have an antagonistic effect, as observed by the production of IFNγ or Il-1B in serum and the ventricular tissue, possibly promoting a pro-inflammatory scenario. Therefore, simultaneous administration of AT-RvD1 and Bz may not be the best strategy. It is uncertain whether this concomitant scheme affects the activity of AT-RvD1. Therefore, it would be more effective to administer a course of AT-RvD1 and resolvin, facilitating efferocytic parasite elimination, followed by a shorter course and/or lower doses of Bz, which would also imply a lower rate of adverse events due to the antiparasitic drug. However, this modality requires further investigation.

The most recurrent lesion in advanced CCC is fibrosis and wall thinning, consistent with dilated cardiomyopathy. Cardiomyocyte loss, due to parasite persistence and the consequent inflammatory process, is characteristic of myocarditis, which is much less intense in indeterminate patients. Here, cardiomyocyte hypertrophy was observed, evidenced by an increase in cell cross-sectional area and correlated with elevated levels of natriuretic peptides, indicating hemodynamic overload. Together with the interstitial fibrosis observed, these findings point to the presence of myocarditis characteristic of CCC [7].

To the best of our knowledge, until now, the effect of AT-RvD1 on hypertrophy produced by infection with *T. cruzi* had not been studied. Herein, AT-RvD1 was shown to reduce both the cross-sectional area of cardiomyocytes and the transcription of hypertrophy markers such as ANP and BNP, which have been associated with different stages and severity of CCC [63,64]. Furthermore, during *T. cruzi* infection, the participation of endothelin-1, cardiotrophin-1, and cytokines such as TNFα and IL-1β has been previously highlighted as pro-

hypertrophic [65]. Therefore, the effect of AT-RvD1 on hypertrophy could also be a consequence of its immunoregulatory effect [66]. Although FPR2 mediates the anti-hypertrophic effect of AT-RvD1, the participation of GPR32 cannot be ruled out. This fact may also explain the changes in ANP and BNP expression observed in the FPR2$^{-/-}$ mice.

Additionally, AT-RvD1 reduced cardiac fibrosis in our experimental model. This finding is consistent with that of a previous report showing that the administration of RvD1 to mice infected with *T. cruzi* reduced cardiac fibrosis and transforming growth factor-β (TGF-β) mRNA levels in the heart [24]. However, we used a 20-day continuous treatment scheme during the early chronic stage of infection instead of the three-bolus scheme used by the previous investigators. In experimental models of myocardial infarction, RvD1 reduced the transcription of profibrotic genes and decreased collagen deposition, thereby reducing fibrosis and improving ventricular function [67]. In a previous study, LXA4 decreased fibroblast proliferation without causing any effect on α-SMA or collagen production in TGF-β1-stimulated fibroblasts [68]. However, in the present model, AT-RvD1 prevented the differentiation of fibroblasts to myofibroblasts, as evidenced by the effect on α-SMA production, which was consistent with decreased collagen production and validated the influence of this pro-resolving lipid on the profibrotic effects of *T. cruzi* infection; this result was in alignment with the findings of Coelho et al [69]. The effect of AT-RvD1 on fibrosis is not dependent on FPR2 exclusively since the antifibrotic effect exerted by AT-RvD1 was not impaired in the FPR2$^{-/-}$ mice. Therefore, we propose that GPR32 may be necessary for this purpose, suggesting a level of redundancy within the resolution of the inflammation cascade.

RvD1 decreased the body mass of *T. cruzi*-infected mice in a previous study [24]; however, in the present study, the mice showed no significant changes in body weight throughout the duration of the experiment (S3 Fig). This discrepancy necessitates further investigation, as it is difficult to explain. The dissimilarity between RvD1 and its epimer could be a possible reason, although speculative.

An electrocardiographic analysis performed on the infected animals showed alterations in the QT interval in the WT mice, suggesting a slowdown in ventricular repolarization. This alteration appears early in patients with CCC and is a mortality predictor in patients with CD [70,71]. Although this QT interval alteration helped confirm the model's chronic nature, AT-RvD1 did not affect this disease pathology at the early stages studied herein. A more prolonged observation period may be necessary to observe significant AT-RvD1-dependent changes in electrical cardiac function. Moreover, the evaluation of cardiac functions, especially electrocardiographic changes, early in the chronic phase may not show significant changes, which could be a limitation for the present study. However, the effect of AT-RvD1 on the pathophysiological events occurring at this stage of the disease, as seen in the changes in hypertrophy and fibrosis, as well as in the levels of biochemical markers of cardiac function, suggests the potential usefulness of host factor modifications in the treatment of CD, especially in the prevention of chronic deterioration arising from inflammatory effects.

## AT-RvD1 reduces parasite load in *T. cruzi*-infected mice

Importantly, AT-RvD1 has no trypanocidal activity. However, we observed that AT-RvD1 alone or combined with Bz reduced the heart parasite load in infected mice. Studies in experimental CCC models have indicated that the administration of 15-epi-LXA$_4$ similarly reduces the parasite load of cardiac tissue in infected mice [25]. In the absence of FPR2, eliminating the parasite from cardiac tissue is not as efficient, suggesting that the pro-resolving cascade is essential for promoting parasite clearance. Probably, the anti-inflammatory effect of AT-RvD1 reduces the deleterious effect of the inflammatory environment that favors parasite

persistence. The pro-resolving processes initiated by AT-RvD1 enable the immune system to eliminate the parasites, probably because chronic inflammation is an evasive mechanism used by the parasites. However, we cannot exclude the fact that the observed clearance is the result of resolvin-induced efferocytosis [72]. Nevertheless, this mechanism implies that the infected cells are close to death or apoptosis, and hence, the mechanisms involved in these processes would be unclear or complicated.

In conclusion, resolution of inflammation may be advantageous as a potential therapeutic strategy for CCC, aiding in the improvement of the current treatment regimens involving antichagasic drugs currently in use. AT-RvD1 is a pro-resolving lipid mediator effective against inflammation, dampening pathological inflammatory responses and restoring tissue homeostasis; importantly, it contributes to clearing adverse inflammatory environments and allowing more efficient parasite elimination, with the participation of FPR2, at least partially. Consequently, drug strategies aimed to modify host factors and resolve inflammation will improve specific antiparasitic drug therapy in the future.

## Supporting information

**S1 Fig. Levels of cytokines in mice infected with Trypanosoma cruzi at 20, 40, and 60 dpi.** The concentrations of TNFα (A-B), IFNγ (C-D), IL-1β (E-F), and IL-10 (G-H) were quantified in the serum from C57BL/6 mice uninfected and infected with T. cruzi at 20, 40, and 60 dpi, using ELISA assays. Data are expressed as the mean ± SEM from one experiment (n = 8 mice per group). Two-way ANOVAs and Tukey's post-hoc tests were performed to identify significant differences. Asterisks indicate significant differences between infected WT and infected FPR2. Dollar signs indicate significant differences between healthy FPR2-/- and infected FPR2-/-. Number signs indicate significant differences between healthy WT and infected WT. one symbol, $p \leq 0.05$; two symbols, $p \leq 0.01$; three symbols, $p \leq 0.001$; four symbols, $p \leq 0.0001$. dpi, days post-infection.
(TIF)

**S2 Fig. Effect of AT-RvD1 and benznidazole in C57BL/6 WT and FPR2-/- mice infected with Trypanosoma cruzi on the cardiac electrical conduction system.** The variation in heart rate (A), QT Interval (B), and QTc (C) are presented. The statistical analysis used was two-way ANOVA followed by Tukey's post-hoc tests (n = 8 mice per group). $^{**}p \leq 0.01$, $^{***}p \leq 0.001$, $^{****}p \leq 0.0001$. QTc, corrected QT interval.
(TIF)

**S3 Fig. Evolution of body weight of mice infected for 60 days with *Trypanosoma cruzi*.** Mice were infected with *T. cruzi* (Dm28) trypomastigotes and treated with 5 μg/Kg//day AT-RvD1 or 30 mg/kg/day Bz between day 40 and 60 postinfection. For the combination, Bz dose was 5 mg/kg/day (n = 8 mice per group)
(TIF)

## Acknowledgments

The authors wish to thank Dr. Jader Dos Santos Cruz from the Department of Biochemistry and Immunology, Institute of Biological Sciences, Federal University of Minas Gerais, Brazil, for performing the mouse electrocardiograms.

## Author Contributions

**Conceptualization:** Ileana Carrillo, Fabiana S. Machado, Guillermo Díaz-Araya, Juan D. Maya.

**Formal analysis:** Ileana Carrillo, Rayane Aparecida Nonato Rabelo.

**Funding acquisition:** Ulrike Kemmerling, Fabiana S. Machado, Guillermo Díaz-Araya, Juan D. Maya.

**Investigation:** Ileana Carrillo, Rayane Aparecida Nonato Rabelo, César Barbosa, Mariana Rates, Sebastián Fuentes-Retamal, Fabiola González-Herrera, Daniela Guzmán-Rivera, Helena Quintero, Ulrike Kemmerling, Christian Castillo.

**Methodology:** Ileana Carrillo, Rayane Aparecida Nonato Rabelo, César Barbosa, Mariana Rates, Sebastián Fuentes-Retamal, Fabiola González-Herrera, Daniela Guzmán-Rivera, Helena Quintero, Christian Castillo, Fabiana S. Machado, Guillermo Díaz-Araya.

**Project administration:** Juan D. Maya.

**Supervision:** Fabiana S. Machado, Guillermo Díaz-Araya, Juan D. Maya.

**Visualization:** Sebastián Fuentes-Retamal, Daniela Guzmán-Rivera, Christian Castillo.

**Writing – original draft:** Ileana Carrillo.

**Writing – review & editing:** Ulrike Kemmerling, Fabiana S. Machado, Guillermo Díaz-Araya, Juan D. Maya.

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
