## [Decision Letter · Decision Letter 0]

13 Sep 2021

Dear Dr Maya,

Thank you very much for submitting your manuscript "Aspirin-triggered resolvin D1 reduces parasitic cardiac load by decreasing inflammation through N-formyl peptide receptor 2 in a chronic murine model of Chagas disease" for consideration at PLOS Neglected Tropical Diseases. As with all papers reviewed by the journal, your manuscript was reviewed by members of the editorial board and by several independent reviewers. In light of the reviews (below this email), we would like to invite the resubmission of a significantly-revised version that takes into account the reviewers' comments. 

The manuscript was evaluated by 3 independent reviewers. One reviewers pointed lack of novelty, which should be carefully addressed by the authors. In their response, please, discuss the papers mentioned by the referee #1 and highlight the novelties that this manuscript brings to the field. Another important concern, pointed by the two other reviewers, is that 60 days can be considered at the most the beginning of chronic phase or the end of acute phase, therefore, a timepoint which "chronic" is debatable. Many of the data shown has already been demonstrated for acute experimental Chagas in the literature. If the authors could provide a further point more into the chronic phase, it would enrich considerably the manuscript. Otherwise, a justification and more precise description of the data is required. The referees also pointed that some of the results are not supported by the data presented. Please, correct all those imprecisions.

We cannot make any decision about publication until we have seen the revised manuscript and your response to the reviewers' comments. Your revised manuscript is also likely to be sent to reviewers for further evaluation.

Sincerely,

Helton da Costa Santiago, M.D., Ph.D

Associate Editor

Brian Weiss

Deputy Editor

The manuscript was evaluated by 3 independent reviewers. One reviewers pointed lack of novelty, which should be carefully addressed by the authors. In their response, please, discuss the papers mentioned by the referee #1 and highlight the novelties that this manuscript brings to the field. Another important concern, pointed by the two other reviewers, is that 60 days can be considered at the most the beginning of chronic phase or the end of acute phase, therefore, a timepoint which "chronic" is debatable. Many of the data shown has already been demonstrated for acute experimental Chagas in the literature. If the authors could provide a further point more into the chronic phase, it would enrich considerably the manuscript. Otherwise, a justification and more precise description of the data is required. The referees also pointed that some of the results are not supported by the data presented. Please, correct all those imprecisions.

Reviewer's Responses to Questions

**Key Review Criteria Required for Acceptance?**

**Methods**

-Are the objectives of the study clearly articulated with a clear testable hypothesis stated?

-Is the study design appropriate to address the stated objectives?

-Is the population clearly described and appropriate for the hypothesis being tested?

-Is the sample size sufficient to ensure adequate power to address the hypothesis being tested?

-Were correct statistical analysis used to support conclusions?

-Are there concerns about ethical or regulatory requirements being met?

Reviewer #1: (No Response)

Reviewer #2: The study goal is sufficiently articulated with a testable hypothesis. The authors attempt to demonstrate that the resolution of inflammation is valuable as a pharmacological strategy for chronic Chagas heart disease. 

Using a murine model, they evaluate administration of AT-RvD1, a pro-resolving lipid mediator effective against inflammation, to dampen pathological inflammatory responses and achieve parasite elimination. Thus, the study design is appropriate to address the stated objective. 

The animal protocol is clearly described and the number of mice included for the different experiments allows validation of the hypothesis being tested. 

Statistical analyses used give support to the authors’ conclusions.

No concerns about ethical or regulatory requirements have been identified. 

Specific comments:

The experimental model is based on C57BL/6 mice at 60 days of infection with T. cruzi strain Dm28c (indicate TcI lineage). This reflects the beginning of prolonged infection, showing moderate inflammatoty pathology and lack of important ECG abnormalities. Therefore, the study title should be changed to “…in a murine model of early chronic Chagas disease”.

On what basis have the authors chosen the dose and duration of treatment with AT-RvD1?

Please indicate the number of independent experiments performed for each figure.

Reviewer #3: -Are the objectives of the study clearly articulated with a clear testable hypothesis stated? YES

-Is the study design appropriate to address the stated objectives? YES

-Is the population clearly described and appropriate for the hypothesis being tested? ANIMAL MODEL

-Is the sample size sufficient to ensure adequate power to address the hypothesis being tested? YES

-Were correct statistical analysis used to support conclusions? YES

-Are there concerns about ethical or regulatory requirements being met? YES

This paper studies the immunomodulatory experimental effects of AT-RvD1 and BZ treatment in a murine model of Chagas disease, isolated and in combination. The most relevant results are the regulation of the inflammatory response both at the systemic level and in the cardiac tissue, and the reduction in cellular infiltrates, cardiomyocyte hypertrophy, fibrosis,

and the parasite load in the heart tissue. The work deserves to be published, but some important issues and limitations should be addressed, especially the following concepts that need to be revised:

Line 35: Both in the “Abstract” and in the “Author Summary” the sentences “After years of infection and in the absence of treatment, the disease progresses from an acute and asymptomatic phase to a chronic inflammatory cardiomyopathy, leading to heart failure and death” and “If the parasite is left untreated, the disease progresses from an acute symptomless phase to chronic 60 myocardial inflammation, which can cause heart failure and death years after infection.” should change to mention the percentage of patients progressing to CCC (30%). In its present form it transmits a misconception that all patients evolve to HF and death, which causes stigma and fear to CD affected people

Line 83- reference in a different format – WHO 2018; reference [1] is WHO 2020

Line 87 – please insert the reference related to 1% of patients treated

Line 91 – Sudden death is NOT the leading cause of mortality in patients with CCC, nor this idea is proposed in reference [5]

Line 95: “The current treatment for CD utilizes nitroheterocyclic drugs, such as benznidazole (Bz), which are far from efficacious. This sentence shows no reference, and this is a concept that is under debate, specially under the results published after comparison of CCC progression rate by BZ treated and non-treated patients in a long-term follow-up cohort (Benznidazole decreases the risk of chronic Chagas disease progression and cardiovascular events: A long-term follow up study - https://doi.org/10.1016/j.eclinm.2020.100694) and the Samitrop study (Beneficial effects of benznidazole in Chagas disease: NIH SaMi-Trop cohort study - 

https://doi.org/10.1371/journal.pntd.0006814). This idea is repeated in the conclusion and should be changed or better contextualized in the manuscript. 

Line 100: “The resolution of inflammation is an active process orchestrated by molecules known as specialized pro-resolving mediators (SPM) that culminate inflammatory processes [8]”. The verb “culminate” does not fit with the idea that authors seem wanting to propose. 

Line 154 – The animal model presents a limitation to be considered as a chronic model for human CCC by day 60 post infection. Two months is too soon to be considered as a chronic model, especially in C57Bl6 mice that are more resistant that Balb/C, CD1 or Swiss mice. Authors should have waited at least 90 -100 days to consider it a chronic model. This might explain why ECG alterations were not detected, as well as significant fibrosis in the WT model. Please inform the number of animals used in each experiment and how many times the experiment was replicated. In the legend of the figures sometimes it is mentioned n=5 or n=8. The results do express a mean of n mice in one experiment or n mice in x experiments? Besides, the authors did not mention if they used male or female mice, or both, a condition that also modulates host response. Dm28c trypomastigotes less virulent than other strains and clones. These issues do not question the quality of the results obtained with the experiments presented but should be considered with reservations concerning interpretations and comparisons with human CCC. 

Line 178: How was the Bz administered orally: in the water or per os?

Line 195: What was the disruption tissue used? 

Line 224: Histology – please inform from what chamber the heart sections were obtained and if it was the same in the mice of all groups.

**Results**

-Does the analysis presented match the analysis plan?

-Are the results clearly and completely presented?

-Are the figures (Tables, Images) of sufficient quality for clarity?

Reviewer #1: (No Response)

Reviewer #2: The analysis presented relatively matches the original aim. The authors state that AT-RvD1 therapy is helpful to revert T. cruzi-elicited proinflammatory cytokine imbalance, mainly via FPR2 (lines 480-482). However, mRNA analysis of heart tissues reveals no significant reduction in myocardial IFNgamma and IL-1beta levels (Fig 3 B,C). Such paragraph should be re-written.

Moreover, AT-RvD1 demonstrated anti-hypertrophic effect through FPR2 in myocardium, but most beneficial actions (decrease in collagen deposition and production of natriuretic peptides; IL-10 overexpression) are independent of interaction with this receptor. Why the study title refers particularly to FPR2?

Additionally, the outcome of coadministration of AT-RvD1 and benznidazole is poorly analyzed. The effects of combination are often worst than those achieved with each individual drug. Does Bz affect AT-RvD1 activity when administered simultaneously? The current findings strongly discourage the potential use of AT-RvD1 + Bz combined therapy for Chagas. A comment about this observation should be included in the Discussion section.

The overall presentation of results is correct, including figures of adequate quality.

Specific comments:

In a recent report by Horta and colleagues (Infect Immun 2020; 88:e00052-20), a decrease in body mass was recorded upon RvD1 treatment in chagasic mice. Have the authors observed such deleterious effect in their series? 

Increased levels of myocardial TGFbeta is a hallmark of T. cruzi-triggered fibrosis. Have the authors any data of this cytokine in AT-RvD1-receiving mice?

Regarding the reduced parasite load in hearts from infected and AT-RvD1-treated mice and, besides, the enhancement of immune effector mechanisms, occurrence of pathogen clearance through efferocytosis should not be discarded, as described previously (Karaji & Sattentau, Front Immunol 2017; 8:1863). This could be commented on within the Discussion.

The alternative receptor for D1 resolvins is GPR 32 (reference 58 of the paper). Please correct line 514 of the text.

Lines 483-485: in addition to NFkappaB inhibition, Gu and colleagues (Innate Immun 2016; 22:186-195) have shown a key role for GSK3beta and CREB in the anti-inflammatory actions of resolvin D1. This finding should be mentioned and the new reference added.

Reviewer #3: -Does the analysis presented match the analysis plan? YES

-Are the results clearly and completely presented? YES, BUT DESERVES SOME CONSIDERATIONS

-Are the figures (Tables, Images) of sufficient quality for clarity? YES, BUT LEGENDS NEED IMPROVEMENTS

Line 267: Fig 1 A. Parasitemia in WT x FPR2 -/- mice infected with Trypanosoma cruzi (Dm28c): Please highlight the 2 peaks, higher at the 2nd at 27th dpi. The parasitemia curves are extremely similar, and only one time point out of 18 shows a statistically difference between the two models. I suggest the authors to be cautious with the conclusion that FPR2 -/- model has lower parasitemia than WT. One single time point and no information of a similar difference in different experiments does not allow such a conclusion. The sentence “but the overall kinetics during the acute phase was similar to that of the WT mice” is not sufficient. 

Lines 298 and 318: In Figs 2 and 3 the legend should be changed. As it is written it seems that AT RvD1 and/or Bz treatment were performed at 60 days post-infection, but in methods it is clearly stated that treatment was conduced from dpi 40 to 60 and immune and histological analysis were done at dpi 60. I suggest: “Fig 2. Serum cytokine levels at 60 days post-infection in mice infected with Trypanosoma cruzi and treated with AT RvD1 and/or Bz in the 20 precedent days.” and a similar format for Fig. 3.

Lines 338, 369, 386, and 408: In Figures 3,5,6, and 7, authors stress “in C57BL/6 chronically infected with Trypanosoma cruzi.” Concerning the limitations of the model as a “chronic” one, I suggest that the legends state “of C57BL/6 mice infected for 60 days with Trypanosoma cruzi.”

**Conclusions**

-Are the conclusions supported by the data presented?

-Are the limitations of analysis clearly described?

-Do the authors discuss how these data can be helpful to advance our understanding of the topic under study?

-Is public health relevance addressed?

Reviewer #1: (No Response)

Reviewer #2: Most conclusions are supported by the data presented (see exceptions detailed above). Some aspects of analysis are limited by the experimental model chosen; e.g. ECG alterations should be better analyzed later in the course of infection. 

The authors do discuss how these data can be helpful to advance our understanding of chronic Chagas heart disease and optimize specific chemotherapy. They clearly explain the worldwide relevance of Chagas cardiomyopathy for public health.

Reviewer #3: -Are the conclusions supported by the data presented? MOST OF THEM. Please see comments regarding the difference in parasitemia

-Are the limitations of analysis clearly described? NO. The authors do not comment on the limitations of their chronic model

-Do the authors discuss how these data can be helpful to advance our understanding of the topic under study? YES

-Is public health relevance addressed? YES

**Editorial and Data Presentation Modifications?**

Reviewer #1: (No Response)

Reviewer #2: Modifications needed to improve the current study have been detailed above.

Reviewer #3: Suggested changes are presented in the other sections

**Summary and General Comments**

Reviewer #1: (No Response)

Reviewer #2: The work by Carrillo and colleagues is a fairly robust and well written study. It displays partial novelty, as the authors use the aspirin- triggered epimer of RvD1, instead of the RvD1 previously tested in mice chronically infected by T. cruzi (Infect Immun 2020; 88:e00052-20). 

An inadequate balance in the inflammatory response is involved in the progression of chronic Chagas cardiomyopathy. Current therapeutic strategies cannot prevent or reverse the heart damage caused by the parasite. In this scenario, the authors evaluated the therapeutic potential of AT-RvD1 as a pro-resolving mediator to arrest heart inflammation in Chagas disease. They found that AT-RvD1 was helpful in reducing leukocyte infiltration, cardiomyocyte hypertrophy, fibrosis, and parasite burden in cardiac tissue, but not to fully restore cytokine balance in the organ. Remarkably, they treated mice early in the course of chronic infection, a model that does not allow analysis of resolvin effects on T. cruzi-dependent heart dysfunction (a phase displaying important ECG disturbances). Further experiments including treatment of mice with more advanced Chagas cardiomyopathy should be required.

Reviewer #3: This paper studies the immunomodulatory experimental effects of AT-RvD1 and BZ treatment in a murine model of Chagas disease, isolated and in combination. The most relevant results are the regulation of the inflammatory response both at the systemic level and in the cardiac tissue, and the reduction in cellular infiltrates, cardiomyocyte hypertrophy, fibrosis,

and the parasite load in the heart tissue. The work deserves to be published, but some important issues and limitations should be addressed, especially the following concepts that need to be revised:

3 main points should be addressed: 

1) the limitation of the model as a "chronic" murine model simiilar to CCC

2) the difference of just one point in parasitemia curves between WT and FPR2 -/- infected mice

3) informations related to BZ treatment limitations

PLOS authors have the option to publish the peer review history of their article (what does this mean?). If published, this will include your full peer review and any attached files.

Reviewer #1: No

Reviewer #2: No

Reviewer #3: No
---

## [Decision Letter · Decision Letter 1]

5 Nov 2021

Dear Dr Maya,

We are pleased to inform you that your manuscript 'Aspirin-triggered resolvin D1 reduces parasitic cardiac load by decreasing inflammation in a murine model of early chronic Chagas disease' has been provisionally accepted for publication in PLOS Neglected Tropical Diseases.

Best regards,

Helton da Costa Santiago, M.D., Ph.D

Associate Editor

Brian Weiss

Deputy Editor

Reviewer's Responses to Questions

**Key Review Criteria Required for Acceptance?**

**Methods**

-Are the objectives of the study clearly articulated with a clear testable hypothesis stated?

-Is the study design appropriate to address the stated objectives?

-Is the population clearly described and appropriate for the hypothesis being tested?

-Is the sample size sufficient to ensure adequate power to address the hypothesis being tested?

-Were correct statistical analysis used to support conclusions?

-Are there concerns about ethical or regulatory requirements being met?

Reviewer #2: (No Response)

**Results**

-Does the analysis presented match the analysis plan?

-Are the results clearly and completely presented?

-Are the figures (Tables, Images) of sufficient quality for clarity?

Reviewer #2: (No Response)

**Conclusions**

-Are the conclusions supported by the data presented?

-Are the limitations of analysis clearly described?

-Do the authors discuss how these data can be helpful to advance our understanding of the topic under study?

-Is public health relevance addressed?

Reviewer #2: (No Response)

**Editorial and Data Presentation Modifications?**

Reviewer #2: (No Response)

**Summary and General Comments**

Reviewer #2: Accept.

PLOS authors have the option to publish the peer review history of their article (what does this mean?). If published, this will include your full peer review and any attached files.

Reviewer #2: No

---

## [Editor Report · Acceptance letter]

11 Nov 2021

Dear Dr Maya,

We are delighted to inform you that your manuscript, "Aspirin-triggered resolvin D1 reduces parasitic cardiac load by decreasing inflammation in a murine model of early chronic Chagas disease," has been formally accepted for publication in PLOS Neglected Tropical Diseases.

Best regards,

Shaden Kamhawi

co-Editor-in-Chief

Paul Brindley

co-Editor-in-Chief
